# AnoFormer: Time Series Anomaly Detection using Transformer-based GAN with Two-Step Masking

## Abstract

Time series anomaly detection is a task that determines whether an unseen signal is normal or abnormal, and it is a crucial function in various real-world applications. Typical approach is to learn normal data representation using generative models, like Generative Adversarial Network (GAN), to discriminate between normal and abnormal signals. Recently, a few studies actively adopt transformer to model time series data, but there is no transformer-based GAN framework for time series anomaly detection. As a pioneer work, we propose a new transformer-based GAN framework, called AnoFormer, and its effective training strategy for better representation learning. Specifically, we improve the detection ability of our model by introducing two-step masking strategies. The first step is *Random masking*: we design a random mask pool to hide parts of the signal randomly. This allows our model to learn the representation of normal data. The second step is *Exclusive and Entropy-based Re-masking*: we propose a novel refinement step to provide feedback to accurately model the exclusive and uncertain parts in the first step. We empirically demonstrate the effectiveness of re-masking step that our model generates more normal-like signals robustly. Extensive experiments on various datasets show that AnoFormer significantly outperforms the state-of-the-art methods in time series anomaly detection.

## 1 Introduction

Time series anomaly detection is a crucial technology to prevent potential risks and financial losses in a variety of areas, such as detecting anomalies on sensor data of large-scale plants [1], ECG monitoring [2], and the network traffic analysis [3]. To deal with this task, from the classic methods [4, 5, 6] to the recent deep learning-based methods [2, 7, 8, 9, 10, 11, 12, 13], many studies have focused on unsupervised learning methods due to the lack of labeled anomalies and highly nonlinear temporal dependencies.

One of major deep learning-based approaches is a reconstruction-based method. It typically uses an autoencoder (AE) or Generative Adversarial Network (GAN) to learn the representation of normal data and to reconstruct a normal-like signal from an input always. As a backbone network, the existing studies widely utilize CNN (Convolutional Neural Networks) [2] or RNN (Recurrent Neural Networks) [8, 9, 10]. More recently, there have been attempts to apply transformer [14] to time series anomaly detection, and it shows remarkable performances [11]. In this work, we also adopt transformer to embed time series representation, but design an adversarial framework for anomaly detection.

If we devise GAN using a transformer encoder, we expect that the model learns normal time series data and eventually generates real normal-like signals. However, there is a major issue. Unlike the AE structure, a pure transformer encoder-based generator does not have a compressed latent space,

*i.e.*, it makes the model find the trivial solution, just copying an input and pasting to the output for the reconstruction. Therefore, we need a new training method for the generator to learn the distribution of normal time series data. To address this issue, we introduce a novel two-step masking strategy. From this approach, the next question is *where to mask an input signal to detect anomalies effectively*. Understandably, in order to make the normal-like output, the best masking positions are abnormal points in the input signal. It is a challenging to mask the abnormal areas selectively because we do not know where the abnormal parts are in advance.

In this paper, we propose AnoFormer, which is a novel transformer-based GAN utilizing a pure transformer encoder only. To learn data representation effectively, we adopt a masking strategy. We first train transformer-based GAN with random masking (Step 1) for representation learning of the normal time series data. While filling the randomly masked parts of the input at Step 1, the model learns the distribution of normal data effectively. In Step 1 alone, all parts of the input signal cannot be considered, and this randomness is a big problem in anomaly detection. Therefore, we solve this problem by re-masking the exclusive parts of Step 1. Also, to find the best masking positions, we calculate entropy from the attention maps of transformer blocks and re-mask the parts with high entropy that is likely to be abnormal points with high uncertainty. This exclusive and entropy-based re-masking (Step 2) provides feedback for better representation learning, eventually improving the anomaly detection performance. We experimentally prove that the proposed two-step masking is essential for AnoFormer to solve anomaly detection problem successfully.

Our contributions can be summarized as follows:

- We propose a simple yet effective transformer-based GAN framework having a generator and a discriminator for unsupervised time series anomaly detection, called AnoFormer. Moreover, we present pre-processing and embedding methods for our framework to deal with time series data effectively.

- We introduce a new two-step masking method to encode the distribution of normal time series data. A newly proposed entropy-based re-masking helps our model to provide the feedback to the uncertain parts based on entropy. From the extensive ablations, we empirically verify that our two-step masking makes our model robust and successfully embed the representation of normal time series data.

- AnoFormer achieves new state-of-the-art results with significant improvements on various unsupervised time series anomaly detection datasets: NeurIPS-TS, MIT-BIH, 2D-gesture, and Power-demand.

## 2 Related Work

Generative models using transformer have been proposed and applied to diverse domains, e.g., computer vision [15, 16, 17, 18], natural language processing [19, 20], and sequence modeling [21, 22]. In particular, these models are used to solve various tasks in the image domain, such as scene generation [15, 16, 23], saliency prediction [18], semantic segmentation [24], and sketch synthesis [25]. Moreover, transformer is presented to solve graph-to-sequence transduction task using graph neural network [26], text generation task [27], and time series forecasting task with the modified self-attention mechanism [28]. We also utilize transformer to construct a generative framework, *i.e.*, having both a generator and a discriminator. In this framework, we propose an appropriate embedding method and loss form to effectively solve the anomaly detection problems.

Many studies have used masking to the transformer architecture for effective representation learning. Including BERT [29], which proposes the Masked Language Model (MLM) technique to pretrain the language representation, many studies also adopt the masking methods, like [30] for action recognition, [30] for text classification task, [31] for text log anomaly detection, [32, 33] for visual representation learning. In [34], the CNN-based model learns the semantic context features by using a multi-scale mask across the whole image with different scales for anomaly detection in image domain. We also use masking in our transformer-based GAN for time series anomaly detection, but unlike the above studies, we propose the two-step masking strategy for training and test to provide feedback that boosts the model to generate the uncertain parts successfully.

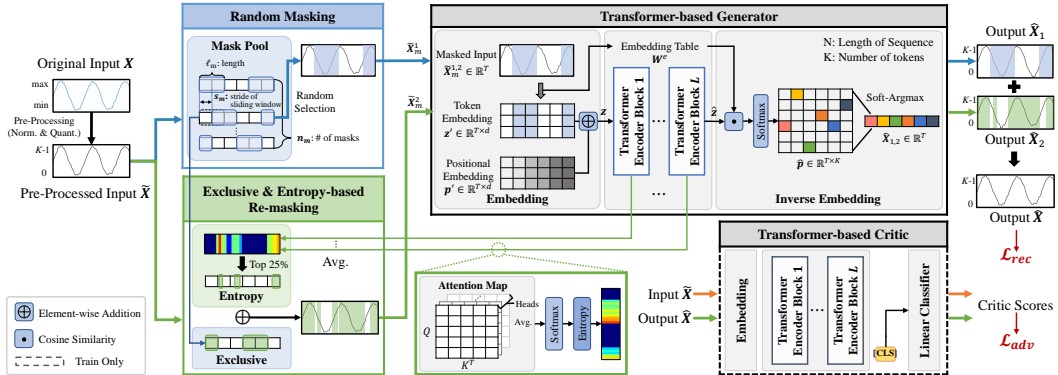

Figure 1: Overview of the proposed AnoFormer. For simplicity, this figure shows the univariate case. In Step 1 (random masking), a pre-processed input $\tilde{X}$ is masked with a randomly selected mask from a predefined mask pool. After passing the masked input $\tilde{X}_m^1$ to the generator, $\hat{X}_1$ is generated as the output by passing through embedding, transformer encoder, and inverse embedding layers. In Step 2 (exclusive and entropy-based re-masking), based on the entropy calculated from attention maps of all layers in Step 1, $\tilde{X}$ is re-masked and $\hat{X}_2$ is generated again from the generator. Final output $\hat{X}$ is constructed via the combination of the masked parts of Step 1 and Step 2. With an aid of a critic, the generator is able to generate more normal-like signals. Here, $\mathcal{L}_{adv}$ is the adversarial loss, including $\mathcal{L}_{adv}^g$ and $\mathcal{L}_{adv}^c$. Note that the critic is used only for the train time.

## 3 AnoFormer

In this section, we propose AnoFormer for unsupervised time series anomaly detection. We first define the target task including an algorithm procedure briefly in Section 3.1. We then describe how to construct a transformer-based GAN framework based on a transformer encoder in Section 3.2. Next, we introduce two different masking steps for our model to encode time series data effectively in Section 3.3. Finally, we present the whole training scheme of AnoFormer in Section 3.4. Figure 1 shows the overall architecture of AnoFormer.

### 3.1 Problem Definition

Let $X = \{x_1, x_2, \cdots, x_T\} \in \mathbb{R}^{T \times n}$ be an input signal of $T$ lengths, where $x_t = \{x_t^1, x_t^2, \cdots, x_t^n\} \in \mathbb{R}^n$ at time step $t$ is a vector of dimension $n$. Since it is easier to get normal time series data compared to abnormal ones, we train a generator G and a discriminator D using only normal data without any label in an unsupervised manner. After training, for each unseen signal $X$, which can be normal or abnormal, the generator $G$ generates a normal-like signal $\hat{X}$. From the generated signal, we can determine whether the observed signal $X$ is normal or not based on the reconstruction errors between the given signal $X$ and the generated signal $\hat{X}$.

### 3.2 Transformer-based GAN for Time Series Data

**Pre-Processing.** To deal with an input signal for a transformer encoder, we need a pre-processing step that makes the input signal discrete tokens. To this end, we normalize each time series input $X$ between -1 and 1 by using the min-max scaling. Then, we quantize the normalized real value within a specific range $[0, K)$, where the integer $K$ is a hyperparameter controlling the quantization resolution, and use the corresponding integer value as a token. Let $\tilde{X} \in \mathbb{R}^{T \times n}$ be the pre-processed signal. We set $K = 400$ for all the experiments, in which the pre-processed signal $\tilde{X}$ looks almost like the input $X$. In total of $K$ tokens (quantization levels), we add a [MASK] token to utilize it for both training and test. To sum up, the input signal $X$ is pre-processed to be $\tilde{X}$ by applying scaling and quantization sequentially.

**Embedding.** An embedding step embeds discrete tokens into the embedding vectors. Here we use a token embedding layer to map each token to the corresponding entry in an embedding weight $W^e \in \mathbb{R}^{(K+1) \times d}$. We denote $z' \in \mathbb{R}^{T \times d}$ as the output token embedding, where $d$ is an embedding

dimension. We add a sinusoidal positional embedding $\boldsymbol{p}'$ to the token embedding $\boldsymbol{z}'$ to allow the model to attend relative positions as follows:

$$\boldsymbol{z} = \boldsymbol{z}' + \boldsymbol{p}'. \tag{1}$$

**Transformer Encoder.** A transformer encoder uses the embedding $\boldsymbol{z} \in \mathbb{R}^{T \times d}$ as the input, and outputs $\hat{\boldsymbol{z}} \in \mathbb{R}^{T \times d}$. Each block of the transformer encoder contains a multi-head self-attention layer and a feed-forward network, followed by a residual connection and a layer normalization. Through the self-attention mechanism, it is possible to attend the relevant information of each time step at once, while multiple attention heads can consider different periodicities in time series data [35].

**Inverse Embedding.** We need to invert the output $\hat{\boldsymbol{z}}$ into the original form of time series, $\hat{\boldsymbol{X}} \in \mathbb{R}^{T \times n}$. To this end, we introduce an inverse embedding layer to our model. We calculate the cosine similarity between the output embedding $\hat{\boldsymbol{z}}$ and the embedding weight $\boldsymbol{W}^e$ taken from the token embedding layer, and apply the softmax operation as follows:

$$\hat{\boldsymbol{p}} = softmax\left( \frac{\hat{\boldsymbol{z}} \cdot \boldsymbol{W}^{e\top}}{\|\hat{\boldsymbol{z}}\| \left\| \boldsymbol{W}^{e\top} \right\|} \right). \tag{2}$$

From the above equation, we obtain the probability distribution $\hat{\boldsymbol{p}} \in \mathbb{R}^{T \times K}$, where $\hat{\boldsymbol{p}}_{t,k}$ means the probability that $k$ will be selected in the range of $[0, K)$ except the [MASK] token at the position $t$. We then extract an index $\hat{\boldsymbol{x}}_t$ of the maximum probability for each time step $t \in [1, 2, \cdots, T]$, using the soft-argmax operation as follows:

$$\hat{\boldsymbol{x}}_t = soft\text{-}argmax(\hat{\boldsymbol{p}}_t) = \sum_{i=0}^{K-1} \frac{e^{\beta \hat{\boldsymbol{p}}_{t,i}}}{\sum_{j=0}^{K-1} e^{\beta \hat{\boldsymbol{p}}_{t,j}}} i,$$

where $\beta$ is a sufficiently large value, such as 1000. Then, the indices in all time steps are concatenated to reconstruct the quantized output $\hat{\boldsymbol{X}}$ as follows:

$$\hat{\boldsymbol{X}} = \{\hat{\boldsymbol{x}}_1, \hat{\boldsymbol{x}}_2, \cdots, \hat{\boldsymbol{x}}_T\}. \tag{3}$$

**Transformer-based GAN Framework.** To enhance the generation quality of $\hat{\boldsymbol{X}}$, we design an adversarial framework using transformer encoders. Following the notation of WGAN-GP [36], from now on we use the term critic $C$ instead of the discriminator $D$. Same as the generator $G$, we construct the critic $C$ using the transformer encoder, but in the critic $C$, a [CLS] token is added in front of the input tokens for classification. After passing through the transformer encoder, the linear classifier outputs the critic score using only the [CLS] token. While classifying the real input $\tilde{\boldsymbol{X}}$ and the fake output $\hat{\boldsymbol{X}}$, the critic $C$ guides the generator $G$ to reconstruct more normal-like signal $\hat{\boldsymbol{X}}$. As a result, our model can distinguish $\tilde{\boldsymbol{X}}$ whether it is normal or abnormal according to the difference between the input signal $\tilde{\boldsymbol{X}}$ and the reconstructed signal $\hat{\boldsymbol{X}}$ from the generator $G$ at test time.

### 3.3 Two-Step Masking for Time Series Encoding

In the previous section, we introduce the transformer-based GAN framework for time series data. However, we empirically find that the representation learning of the proposed transformer-based GAN is not possible because the generator $G$ just copies the input as the output always. Inspired by recent studies [35, 32, 33] that effectively learn the representation through masking in transformer, we propose two different masking steps during training and test time: 1) random masking and 2) exclusive and entropy-based re-masking. We experimentally demonstrate that the proposed two-step masking is essential for our framework to learn the distribution of normal time series data successfully. In the following content, we describe how to mask the input effectively in each step with details.

**Step 1: Random Masking.** As the first step, we partially hide the input signal $\tilde{\boldsymbol{X}}$ using a randomly selected mask from a mask pool. To construct the mask pool, we design a single mask in which the mask and the non-mask sections alternately appear. We then generate multiple masks by applying sliding window to the single mask, and group them as the mask pool. The composition of the mask pool depends on a length $l_m$ of a single mask section, a ratio $r_m$ of all mask parts, and a stride $s_m$ for the sliding window. The number of masks $n_m$ in the predefined mask pool is determined as follows:

$$n_m = 2 \times \left\lceil \frac{l_m}{s_m} \right\rceil. \tag{4}$$

Using the above equation, we generate the enough number of masks in the pool to cover all sections of the signal. During the train and test time, the mask is randomly selected in the predefined mask pool per each signal, and the generator $G$ reconstructs $\hat{\boldsymbol{X}}_1$ from the masked input $\tilde{\boldsymbol{X}}_m^1$.

**Step 2: Exclusive and Entropy-based Re-Masking.** After $\hat{\boldsymbol{X}}_1$ is generated from Step 1, we again mask the exclusive parts that are not covered in Step 1 for our model to consider all parts of the input. To avoid the error accumulation, here we re-mask the input $\tilde{\boldsymbol{X}}$, instead of the first output $\hat{\boldsymbol{X}}_1$. In addition, we provide feedback to our model by re-masking the parts that the model considers uncertain during Step 1. To this end, we get an attention map from each layer of the generator as follows:

$$A^{l,h} = softmax\left(\frac{Q^h K^{h^T}}{\sqrt{d}}\right),$$

$$A^l = \frac{1}{H}\sum_{h=1}^{H} A^{l,h},$$

where $l \in [1, 2, \cdots, L]$ and $A^l$ is the attention map in the $l$-th layer, calculated by the average of all attention maps for individual heads, $A^{l,h}$. This layer-wise attention map determines how much a specific time step focuses on the other parts of the input per signal. In this context, the uniformly distributed attention means that the model does not know which connections are valuable [37], *i.e.*, the prediction is uncertain. To quantify the uncertainty, we calculate an entropy $H_{\hat{\boldsymbol{X}}_1}$ of the masked input $\hat{\boldsymbol{X}}_1$ as follows:

$$H(t) = -\frac{1}{L}\sum_{l=1}^{L}\sum_{j=1}^{T} A_{t,j}^l \log A_{t,j}^l,$$

$$H_{\hat{\boldsymbol{X}}_1} = \{H(1), H(2), \cdots, H(T)\}.$$

To provide feedback on the high entropy parts, we re-mask 50% of the parts already masked in Step 1. Then the generator $G$ re-generates the second output $\hat{\boldsymbol{X}}_2$ from the masked signal $\tilde{\boldsymbol{X}}_2$. Finally, we combine the masked parts generated from Step 1 and the ones from Step 2 to construct the final output $\hat{\boldsymbol{X}}$. If there are overlapped parts between Step 1 and Step 2, the parts of Step 2 are used. From this re-masking step, we experimentally prove that our model becomes robust to unexplored and uncertain parts within a fixed model size. We also use the same random masking and re-masking strategies at test time.

## 3.4 Training AnoFormer

To train AnoFormer, we apply the cross-entropy loss to reconstruct the same input $\tilde{\boldsymbol{X}}$ from the final output $\hat{\boldsymbol{X}}$ as follows:

$$\mathcal{L}_{rec} = -\sum_{i=1}^{T}\sum_{j=1}^{K} \tilde{\boldsymbol{X}}_{i,j} \cdot \log(\hat{\boldsymbol{p}}_{i,j}), \tag{5}$$

where $\tilde{\boldsymbol{X}}_{i,j}$ denotes the one-hot label vector from the input and $\hat{\boldsymbol{p}}_{i,j}$ denotes the probability distribution of the final output $\hat{\boldsymbol{X}}$. Using $\hat{\boldsymbol{X}}$ from the generator $G$ during two-step masking, the critic $C$ tries to minimize the following loss function:

$$\boldsymbol{X}' = \epsilon\tilde{\boldsymbol{X}} + (1-\epsilon)\hat{\boldsymbol{X}}, \tag{6}$$

$$\mathcal{L}_{C,adv} = \left(\mathbb{E}\left[C\left(\hat{\boldsymbol{X}}\right)\right] - \mathbb{E}\left[C\left(\tilde{\boldsymbol{X}}\right)\right]\right) + \lambda\mathbb{E}_{\boldsymbol{X}'\sim P_{\boldsymbol{X}'}}\left[(\|\nabla_{\boldsymbol{X}'}C\left(\boldsymbol{X}'\right)\|_2 - 1)^2\right], \tag{7}$$

where $\epsilon$ is randomly chosen between zero and one. The first term measures the Wasserstein distance and the second term is the gradient penalty, where $\boldsymbol{X}'$ is a random sample from $P_{\boldsymbol{X}'}$ to enforce the Lipschitz constraint. The coefficient is a harmonic parameter to balance the Wasserstein distance and the gradient penalty, where we use the value of 10. The loss function of the generator $G$ is as follows:

$$\mathcal{L}_{adv}^g = -\mathbb{E}\left[C\left(\hat{\boldsymbol{X}}\right)\right], \tag{8}$$

Table 1: Quantitative comparisons in four datasets. For all of the metrics, a higher value indicates a better performance.

| Metric | Base Architecture | Method | NeurIPS-TS | | | | | | MIT-BIH | 2D-gesture | Power-demand |
|---|---|---|---|---|---|---|---|---|---|---|---|
| | | | Global | Contextual | Shapelet | Seasonal | Trend | Average | | | |
| AUROC | CNN | BeatGAN | 0.9753 | 0.6128 | 0.7398 | 0.9742 | **1.0000** | 0.8372 | *0.9475* | 0.7256 | 0.5796 |
| | RNN | TadGAN | *1.0000* | 0.4285 | *0.9834* | *0.9744* | 0.9327 | *0.9726* | 0.8256 | 0.5294 | *0.8438* |
| | | RAE-ensemble | 0.5226 | *0.9348* | 0.9244 | 0.9625 | 0.7246 | 0.8138 | - | 0.7808 | 0.6587 |
| | | RAMED | 0.5265 | 0.9325 | 0.9084 | 0.9628 | 0.7259 | 0.8112 | - | 0.7839 | 0.6787 |
| | Transformer | Anomaly Transformer | 0.9931 | 0.6224 | 0.7407 | 0.9332 | 0.9976 | 0.8400 | 0.8108 | *0.7868* | 0.7739 |
| | | **AnoFormer (Ours)** | **1.0000** | **0.9758** | **0.9900** | **0.9985** | *0.9985* | **0.9911** | **0.9552** | **0.8407** | **0.8667** |
| AUPRC | CNN | BeatGAN | 0.9855 | 0.7051 | 0.6817 | 0.9748 | **1.0000** | 0.9634 | *0.9143* | 0.4952 | 0.1228 |
| | RNN | TadGAN | *1.0000* | 0.3603 | *0.9565* | *0.9754* | 0.8731 | *0.9806* | 0.4621 | 0.4367 | 0.3098 |
| | | RAE-ensemble | 0.0453 | *0.8297* | 0.8159 | 0.9191 | 0.1378 | 0.5496 | - | 0.5287 | 0.1400 |
| | | RAMED | 0.0443 | 0.8223 | 0.6873 | 0.9109 | 0.1291 | 0.5188 | - | 0.5331 | 0.1627 |
| | Transformer | Anomaly Transformer | 0.9959 | 0.6957 | 0.6630 | 0.9364 | 0.9978 | 0.9639 | 0.5603 | *0.5607* | *0.4967* |
| | | **AnoFormer (Ours)** | **1.0000** | **0.9854** | **0.9901** | **0.9985** | *0.9987* | **0.9982** | **0.9187** | **0.6142** | **0.5584** |
| F1 score | CNN | BeatGAN | 0.9345 | 0.7348 | 0.6136 | 0.9487 | **1.0000** | 0.9008 | *0.8015* | 0.4941 | 0.2266 |
| | RNN | TadGAN | *1.0000* | 0.3590 | *0.9331* | *0.9844* | 0.8170 | *0.9380* | 0.5289 | 0.4138 | 0.5714 |
| | | RAE-ensemble | 0.0853 | *0.8343* | 0.7750 | 0.9181 | 0.3889 | 0.6003 | - | 0.5511 | 0.2678 |
| | | RAMED | 0.0838 | 0.8272 | 0.6203 | 0.8782 | 0.4040 | 0.5627 | - | 0.5633 | 0.2934 |
| | Transformer | Anomaly Transformer | 0.9751 | 0.7358 | 0.6115 | 0.8730 | 0.9958 | 0.9014 | 0.5446 | *0.6486* | *0.6053* |
| | | **AnoFormer (Ours)** | **1.0000** | **0.9400** | **0.9696** | **0.9913** | *0.9974* | **0.9798** | **0.8410** | **0.6667** | **0.6226** |

which makes the critic $C$ not be able to classify the generated $\hat{X}$. To sum up, the proposed AnoFormer is trained via the following loss functions for the generator $G$ and the critic $C$:

$$\mathcal{L}_G = \lambda_{rec}\mathcal{L}_{rec} + \lambda_{adv}\mathcal{L}^g_{adv}, \tag{9}$$

$$\mathcal{L}_C = \mathcal{L}^c_{adv}, \tag{10}$$

where we set $\lambda_{rec}$ and $\lambda_{adv}$ as 1.

# 4 Experiments

**Datasets.** We evaluated AnoFormer on four real-world benchmarks: 1) MIT-BIH [1] contains 48 ECG records of test subjects from Beth Israel Hospital, 2) 2D-gesture contains time series of X and Y coordinates of an actor's right hand, 3) Power-demand is a dataset measuring the power comsumption for the Dutch research facility, and 4) NeurIPS-TS [2] [38] is a synthetic dataset including five different time series anomaly scenarios as point-global, point-contextual, pattern-shapelet, pattern-seasonal, and pattern-trend. More details on each dataset are summarized in Appendix A.

**Baselines.** We compared our model with various baselines, including CNN, RNN, and transformer-based reconstruction models. BeatGAN [2] and TadGAN [8] are CNN and LSTM-based GAN models, respectively. RAE-ensemble [9] is an ensemble of RNNs with sparse skip connections in autoencoder. RAMED [10] additionally uses the multiresolution decoding based on RAE-ensemble. Anomaly Transformer [11] develops the transformer architecture to utilize association information.

**Implementation Details.** For both the generator $G$ and the critic $C$, we utilized the basic transformer encoders with 9 and 6 layers for MIT-BIH, and 4 and 2 layers for other datasets, respectively. The embedding dimension and the number of heads are 128 and 8, respectively. The mask length $l_m$ is about 10% of the sequence length $T$, and the mask stride $s_m$ is about half of the mask length $l_m$. We used Adam optimizer with initial learning rate, momentum $\beta_1$, and $\beta_2$ as 0.0001, 0.5, and 0.999, respectively. We implemented our model using PyTorch and trained on a NVIDIA RTX 3090 GPU.

## 4.1 Quantitative Results

Table 1 shows the anomaly detection performances of each baseline on three different real-world datasets (*i.e.*, MIT-BIH, 2D-gesture, and Power-demand), and a synthetic dataset (*i.e.*, NeurIPS-TS [38]). Overall, RNN or transformer-based models showed high performances except MIT-BIH. In MIT-BIH, BeatGAN showed the second-best performances among all the benchmarks. In case of the proposed AnoFormer, this model outperformed all the baselines in four different datasets. In

[1] https://physionet.org/content/mitdb/1.0.0/
[2] https://github.com/datamllab/tods/tree/benchmark

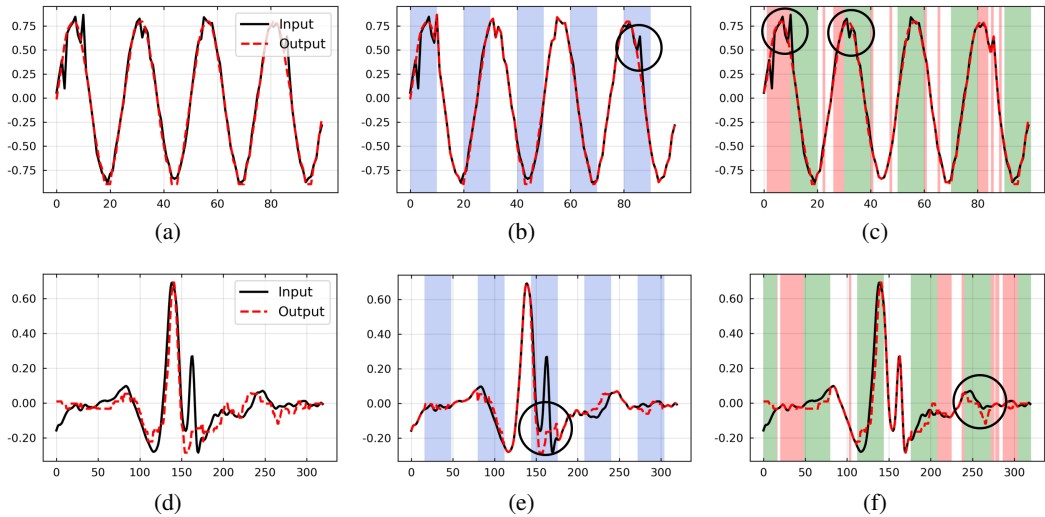

Figure 2: Output visualization in Point-Contextual (NeurIPS-TS) and MIT-BIH datasets. Left: visualization of abnormal input and the normal-like output. Middle: reconstruction results of random masking (blue). Right: reconstruction results of exclusive (green) and entropy-based (red) re-masking. Vest viewed in color.

particular, our model performed well on NeurIPS-TS containing five types of outliers, and it means AnoFormer is robust to the various types of outliers. AnoFormer achieved the state-of-the-art results from small datasets (*e.g.*, 2D-gesture and Power-demand) with about 1,000 training sets to large datasets (*e.g.*, NeurIPS-TS and MIT-BIH) with about tens of thousands of training sets, and from univariate to multivariate cases. The experimental results demonstrate that the proposed transformer-based GAN framework with the two-step masking strategy is effective to reconstruct normal time series data for anomaly detection.

## 4.2 Qualitative Results

Figure 2 shows the qualitative examples of AnoFormer. First column (Figure 2(a) and Figure 2(d)) shows the abnormal examples of point-contextual of NeurIPS-TS and MIT-BIH datasets, respectively. The other columns show that the proposed rnadom masking and re-masking strategies actually provide feedback to our framework. For example, the incorrectly copied parts in Step 1 was refined by the entropy-based re-masking (please see the black circles in the figure). As shown in the figure, when the abnormal inputs were received, the model generated the normal-like outputs. Therefore, AnoFormer can detect the abnormal points through the difference between the input and the output.

## 4.3 Ablation Study

We conducted various ablation studies to analyze the effectiveness of the proposed transformer-based GAN framework and two-step masking. All of the ablation studies were performed on the Point-Contextual dataset of NeurIPS-TS, since it is the most difficult task to detect the anomalies out of the five types of outliers. Figure 2 shows an example of Point-Contextual dataset, which has the small glitches as the outliers. In Appendix B, we additionally examined the sensitivity of each hyperparameter newly adopted in our model.

### 4.3.1 Transformer-based GAN Framework

We first investigated the effectiveness of the transformer-based adversarial framework in our model. In this experiment, we used BeatGAN as a CNN-based baseline. Table 2 shows the ablation results when the generator and the critic use different backbone networks, such as CNN, and transformer. As shown in the table, the transformer-based generator showed higher performances on all of metrics with large margins than the CNN-based generator. Interestingly, we empirically found that there was no synergy when using CNN-based critic with the transformer-based generator. It means, it is not helpful for the transformer-based generator to construct the critic with an inappropriate baseline. On

Table 2: Ablation study of the proposed transformer-based GAN.

| Generator | Critic | AUROC | AUPRC | F1 score |
|---|---|---|---|---|
| CNN | CNN | 0.6128 | 0.7051 | 0.7348 |
| Transformer | - | 0.9572 | 0.9735 | 0.9093 |
| Transformer | CNN | 0.9510 | 0.9675 | 0.9026 |
| **Transformer** | **Transformer** | **0.9758** | **0.9854** | **0.9400** |

Table 3: Ablation study of the proposed two-step masking.

| Step 1 | Step 2 | AUROC | AUPRC | F1 score |
|---|---|---|---|---|
| - | - | 0.5000 | 0.3602 | 0.2386 |
| Random | - | 0.8557 | 0.7959 | 0.7548 |
| Mask pool | - | 0.9109 | 0.8651 | 0.8200 |
| Mask pool | Mask pool (50%) | 0.9277 | 0.8590 | 0.7808 |
| Mask pool | Exclusive (50%) | 0.9489 | 0.9622 | 0.9057 |
| Mask pool | Exclusive + Random (75%) | 0.9709 | 0.9466 | 0.9004 |
| Mask pool | Exclusive + Anomaly score (75%) | 0.9747 | 0.9533 | 0.9119 |
| **Mask pool** | **Exclusive + Entropy (75%)** | **0.9758** | **0.9854** | **0.9400** |

the other hand, the transformer-based critic showed better performances than the baseline without the critic, which means it encourages the generated output to be close to the normal signal. From this result, we demonstrate that our transformer-based GAN framework trained with the proposed masking strategy is effective to reconstruct normal time series data for anomaly detection.

### 4.3.2 Two-Step Masking

As shown in Table 3, we investigated the effect of two-step masking in our model. The first row means a naive form of the transformer-based GAN without any masking. The result was 0.5 of AUROC, which means the naive transformer-based GAN cannot distinguish between normal and abnormal signals at all. To overcome this critical issue, we adopted various masking strategies. First, we investigated the masking for Step 1. Here, *Random* means a fully random masking without any predefined mask pool. *Mask Pool* means our predefined mask pool defined in Section 3.3. The results showed that regardless of the masking strategy, masking itself during training and test enabled the transformer-based GAN to effectively learn the distribution of normal time series data. Moreover, we confirmed that *Mask Pool* is much better than *Random* masking, because each mask in the mask pool definitely covers the different parts from each other, providing a complementary effect.

Next, we conducted in-depth experiments to evaluate and compare different re-masking strategies in Step 2. *Mask pool* method in Step 1 can be also used for re-masking. *Exclusive* method re-masks the exclusive parts of the random mask selected in Step 1. From the results, we found that re-masking improved the detection ability of our model, and especially, exclusive masking strategy was really effective. This is because the model can consider the characteristic of whole signal during two-step masking. To provide more feedback to our model, we additionally re-masked the masked parts in Step 1. We experimented the following three cases: 1) *Random* method re-masks the signal randomly, 2) *Anomaly score* method re-masks the parts with high anomaly scores, and 3) *Entropy* method re-masks the parts with high entropy values. The results showed that masking the uncertain parts provided the proper feedback to our model, resulting in the highest scores among all the baselines. Therefore, we confirmed that the entropy-based re-masking is more effective than the other additional masking methods.

## 5 Discussion

We further conducted analysis to demonstrate the effectiveness of the proposed two-step masking strategy. To confirm the importance of Step 2, we compared our method with the absence of Step 2. Here, we also used Point-Contextual dataset in NeurIPS-TS for analysis.

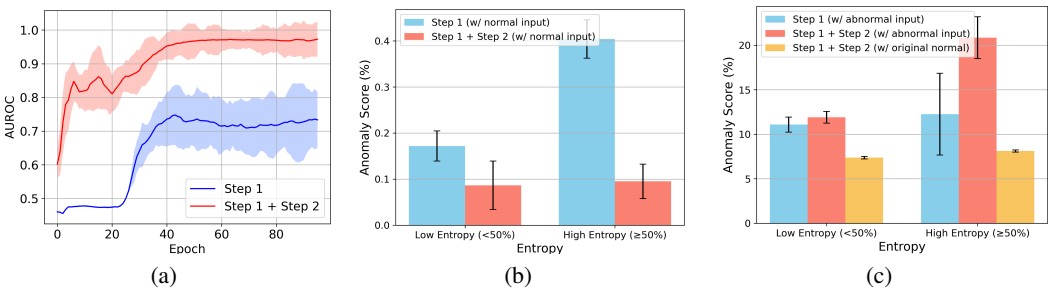

(a)               (b)               (c)

Figure 3: (a) AUROC and std of all masks in the predefined mask pool. (b) Analysis on the entropy-based re-masking strategy (normal-case). (c) Analysis on the entropy-based re-masking strategy (abnormal-case).

**The Effectiveness of Re-Masking.** First, we investigated the validity of re-masking step. We reported the average AUROC and std for all masks in the predefined mask pool in Figure 3. In Step 1, there was a problem that the standard deviation (std) was too high because both training and test time had randomness in selecting the mask parts. By re-masking through Step 2, the randomness of the masked parts was eliminated and the std was significantly reduced. The performance of anomaly detection also increased with a large margin by referring to the entire signal.

**Analysis on Entropy-based Re-Masking.** To understand the effectiveness of entropy-based re-masking intuitively, we visualized the relation between entropy and anomaly score in Figure 3(b) and Figure 3(c) for both cases of normal and abnormal. Since the entropy-based method re-masked the parts selected in Step 1, we measured the anomaly score only in the parts corresponding to Step 1. We found two meaningful insights through the analysis. First, *the higher the entropy, the more incorrect signal the model generates.* In training phase, the high anomaly score means that the model generates output incorrectly, because only normal data is used. From the result of Figure 3(b), the entropy was also high in the parts with high anomaly score, which means that the model did not generate signals well in the parts with high entropy during Step 1. This is because in the high entropy the attention is uniformly distributed and the meaningful connection is not learned. By re-masking these parts in Step 2, anomaly score was significantly reduces, which means that the model reconstructed the normal data well in the training process. Second, *the entropy-based masking improves the discriminative ability between normal and abnormal in test time.* As shown in Figure 3(c), likewise in the case of normal, the higher the entropy, the higher the anomaly score in abnormal case. However, there was also a part with a high entropy and a low anomaly score. These parts mean that the model copied the abnormal input as it was without making it normal. It is possible to provide feedback in both cases with entropy-based re-masking. By re-masking the parts with high entropy, anomaly score was considerably increased, and it means the parts that were not well generated due to copying in Step 1 were well re-generated. This makes it possible to further discriminate between normal and abnormal through anomaly score. In fact, a large anomaly score does not mean getting close to normal data. We used a NeurIPS-TS dataset to see if the output gets closer to normal data through re-masking. Since we synthesized abnormal datasets by injecting sporadic outliers in an additive manner, we could easily get the original normal version of the abnormal data. From this, we confirmed that the output was correctly getting closer to the original normal through Step 2. We further experimented about *which layer's entropy information should be used?* From the results in Appendix C, we calculated entropy from all layers and averaged them.

# 6 Conclusion

In this paper, we introduce AnoFormer, a novel transformer-based GAN for time series anomaly detection. To learn time series data directly with our model, we propose pre-processing and embedding methods suitable for time series data. A new training scheme based on two-step masking enables AnoFormer to embed the representation of normal signals. Especially, the exclusive and entropy-based re-masking method significantly improves the anomaly detection performances on several benchmark datasets. From the extensive experiments, we empirically demonstrate that our model is really effective to solve time series anomaly detection. As future work, we plan to study novel techniques for shorter inference time, and deal with time series data longer than an hour or a day.

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
