# Supplementary:
# AnoFormer: Time Series Anomaly Detection using Transformer-based GAN with Two-Step Masking

**Anonymous Author(s)**
Affiliation
Address
email

Here we provide a brief outline of the appendices. In Appendix A, we provide details of datasets. In Appendix B, we provide additional experiments about hyperparameter sensitivity. In Appendix C, we discuss how to select the layer for entropy-based re-masking strategy. In Appendix D, we study the effectiveness of our embedding method. In Appendix E and F, we discuss the limitations and broader impacts of our methods.

## A   Dataset

We used a total of four time series anomaly detection datasets. In the training set, all datasets contain normal data only. Table 1 shows details of each dataset.

1. NeurIPS-TS[1]: This is the dataset which introduces a new taxonomy for time series outliers. It includes total of five different time series anomaly scenarios that cover point-global, point-contextual, pattern-shapelet, pattern-seasonal, and pattern-trend. We created our own dataset using the open source code. Datasets will be made public after the review.

2. MIT-BIH Arrhythmia Database[2]: This database contains 48 ECG records of test subjects from Beth Israel Hospital. As recommended by the Association for the Advancement of Medical Instrumentation (AAMI) [1], there are five classes that are Normal (N), Supraventricular Ectopic Beat (S), Ventricular Ectopic Beat (V), Fusion (F), and Unknown Beat (Q).

3. 2D-gesture [3]: This dataset contains time series of X and Y coordinates of an actor's right hand. The actor grabs a gun from his hip-mounted holster, and then shoots at the target. Finally, the actor returns it to the holster. The anomalous region is that the actor misses the holster when returning the gun.

4. Power-demand [4]: This is the dataset measuring the power consumption for the Dutch research facility for the entire year of 1997.

## B   Hyperparameter Sensitivity

For the proposed AnoFormer, we set the quantization resolution $K$ as 400 in the main paper. Moreover, for the proposed two-step masking, we set the mask ratio $r_m$ as 50%, the mask length $l_m$ as 10% of the sequence length, the stride of sliding window as a half of $l_m$, and the re-masking ratio as 50% of the parts masked in Step 1. We further analyzed the sensitivity of hyperparameters in AnoFormer.

---

[1]https://github.com/datamllab/tods/tree/benchmark
[2]https://physionet.org/content/mitdb/1.0.0/,
[3]https://www.cs.ucr.edu/ eamonn/discords/.
[4]https://www.cs.ucr.edu/ eamonn/discords/.

Submitted to 36th Conference on Neural Information Processing Systems (NeurIPS 2022). Do not distribute.

Table 1: Statistical details of four datasets.

| Datasets | | Dimension | Length | # Training | # Validation | # Test |
|---|---|---|---|---|---|---|
| NeurIPS-TS | (A) Point-Global | 1 | 100 | 18,000 | 8,954 | 11,927 |
| | (B) Point-Contextual | 1 | 100 | 18,000 | 8,954 | 11,927 |
| | (C) Pattern-Shapelet | 1 | 100 | 18,000 | 8,954 | 11,927 |
| | (D) Pattern-Seasonal | 1 | 100 | 18,000 | 8,954 | 11,927 |
| | (E) Pattern-Trend | 1 | 100 | 18,000 | 8,954 | 11,927 |
| MIT-BIH | | 1 | 320 | 62,436 | 8,025 | 27,107 |
| 2D-gesture | | 2 | 64 | 1,093 | 469 | 46 |
| Power-demand | | 1 | 512 | 1,088 | 467 | 224 |

First, we provide the performances and the visualization results for the different number of $K$ in Table 2. We found that our model is not sensitive to the different values of $K$, and we set $K = 400$ consistently for all experiments because this is visually similar to the original signal.

Table 2: Sensitivity for the quantization resolution.

| # of tokens | 100 | 200 | 300 | 400 | 500 |
|---|---|---|---|---|---|
| AUROC | 0.9791 | 0.9806 | 0.9793 | 0.9758 | 0.9719 |
| AUPRC | 0.9880 | 0.9883 | 0.9874 | 0.9854 | 0.9826 |
| F1 score | 0.9453 | 0.9435 | 0.9388 | 0.9400 | 0.9289 |

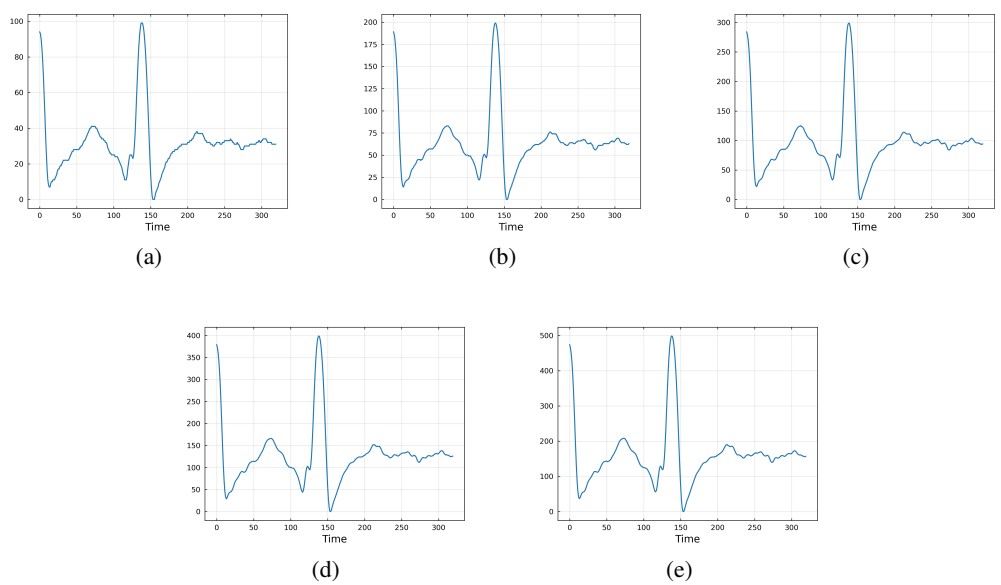

Figure 1: Visualization for each $K$ in MIT-BIH dataset: (a) $K = 100$, (b) $K = 200$, (c) $K$=300, (d) $K = 400$, and (e) $K = 500$. When $K$ is 100 or 200, those signals look discrete ones.

Table 3 shows the performances for the different strides of sliding window $s_m$ while fixing the length of the mask section $l_m$ as 10% of $T$. As already explained, the number of masks $n_m$ in the mask pool was automatically determined by in Equation 1 by using the values of $s_m$ and $l_m$. Our model

Table 3: Sensitivity for the stride of the sliding window.

| Stride | 3 | **5** | 10 |
|---|---|---|---|
| # of pool | 8 | **4** | 2 |
| AUROC | 0.9714 | **0.9758** | 0.9721 |
| AUPRC | 0.9828 | **0.9854** | 0.9834 |
| F1 score | 0.9344 | **0.9400** | 0.9315 |

performed better when the stride was 5, which is a half of $l_m$.

$$n_m = 2 \times \left\lceil \frac{l_m}{s_m} \right\rceil. \tag{1}$$

We also present the compared results depending on the different re-masking ratios in Table 4. When we re-masked a half of the masked parts in Step 1 for the proposed entropy-based re-masking, the performance was the highest among them. To sum up, AnoFormer shows the robust performances to the changes of the hyperparameters.

Table 4: Sensitivity for the entropy-based re-masking ratio.

| entropy-based re-masking ratio | +10% | +20% | **+25%** | +30% | +40% |
|---|---|---|---|---|---|
| AUROC | 0.9682 | 0.9726 | **0.9758** | 0.9615 | 0.9285 |
| AUPRC | 0.9797 | 0.9841 | **0.9854** | 0.9747 | 0.9538 |
| F1 score | 0.9291 | 0.9363 | **0.9400** | 0.9115 | 0.8823 |

## C  How to Select the Layers for Entropy-based Re-masking?

For entropy-based re-masking, we average the entropies from all layers. Attentions are often uniformly distributed in the first block of Transformer [2]. We further analyzed the performance of the last-layer usage. This is because the last layer reflects the characteristics of the data the best. As shown in Table 5, using all layers for re-masking achieved the best performance. This results show that each layer of transformer contains meaningful information to reconstruct the signal.

Table 5: Experiment to select layers to calculate entropy.

| Layer Selection | AUROC | AUPRC | F1 score |
|---|---|---|---|
| Last layer | 0.9735 | 0.9846 | 0.9371 |
| **All layers** | **0.9758** | **0.9854** | **0.9400** |

## D  Effectiveness of the Proposed Embedding Method

The existing transformer-based time series processing methods utilize linear layers for token embedding and use the mean squared error to reconstruct time series data. Different from these studies, we apply the embedding matrix to process time series data. To this end, we replace the reconstruction loss to the cross-entropy. In other words, we change the regression problem to the simple classification one. We further examine the effects of the proposed embedding method. As shown in Figure 2, using embedding matrix achieved better performance with a large margin and converged quickly. We empirically demonstrated that the proposed embedding strategy is effective to process time series data and is superior to the reconstruction-based anomaly detection problem.

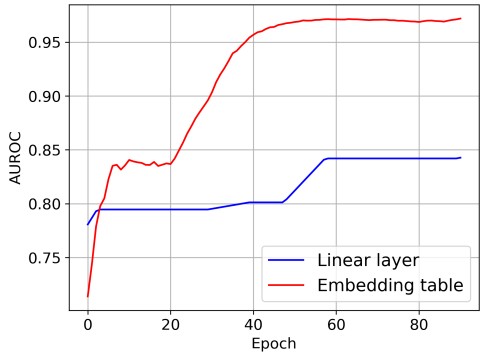

Figure 2: Performance comparison of different embedding methods.

## E    Limitation

Since we combine the signals generated through Step 1 and Step 2 for the final output, there is a problem of noisiness at the mask boundary. In addition, in Step 2 of the proposed masking method, we provide entropy-based feedback to the masked part in Step 1, but we do not provide feedback on the exclusive part of Step 2. Moreover, there is a problem that the inference time is long because we perform the forwarding operation twice for one signal. However, this is not a big deal because accuracy is more important than detecting in real time. In the future, we plan to solve those problems.

## F    Broader Impact

Recently, many researches actively conduct anomaly detection using deep learning. Accordingly, not only does the performance improve, but many industrial areas apply anomaly detection model effectively in the real world. Anomaly detection is generally used positively. It improves safety and prevents potential risks and financial losses by detecting anomalies in healthcare, manufacturing, and autonomous driving, etc. However, the system can be stuck into confirmation bias, *i.e.*, the model can ignore new forms of anomalies. We can cope with this situation by updating the model periodically for new knowledge.