# OpenReview forum: "AnoFormer: Time Series Anomaly Detection using Transformer-based GAN with Two-Step Masking"
_NeurIPS.cc/2022/Conference — NeurIPS 2022 Submitted_

### Official Review · Reviewer_73J5 · 2022-07-07

**Rating:** 3
**Confidence:** 4
**Soundness:** 2 fair
**Presentation:** 4 excellent
**Contribution:** 2 fair

**Summary:**

This paper tackles the task of detecting and localizing anomalies in time series data. It follows a transformer approach where time series are automatically masked, then the transformer attempt to reconstruct the masked region. The reconstruction error is used as an anomaly score. A few technical tricks are incorporated such as using a GAN loss in training and an adaptive masking strategy. The method is evaluated on a few datasets and achieves better results than a set of deep learning approaches.

**Questions:**

* How were the benchmark datasets chosen?
* How do the classical benchmarks for NeurIPS-TS/[2] etc. perform?
* How well does the method scale to multivariate TS (e.g. 100 dimensions)?



**Limitations:**

Some were discussed in the appendix. A few more were highlighted in the review. A more extensive discussion would have been helpful.

**Strengths And Weaknesses:**

Originality: The main premise of using inpainting as a task for reconstruction-based anomaly detection in not new (e.g. [1] but there are many). Nor is using a GAN discriminator (also found in [1]). The main difference here is using a transformer, the particular masking approach and the particular hyperparameter choices. This is an original contribution but a modest one.

Quality: The method appears sound as far as it goes. The GAN ablation is somewhat worrisome as it appears to be useful only in a particular configuration and could be due to randomness in the data or a particular hyperparameter choice. However, the evaluation is not very convincing. The NeurIPS-TS datasets were generated by the authors, and are really easy. Note the original paper had classical baselines as doing much better than deep learning methods - but the original datasets were much smaller than the ones generated here. Also the other 3 benchmarks are not standard. It is not clear to me how they were selected. Also simple/classical baselines were not attempted. Several papers (e.g. NeurIPS-TS cited in this paper or [2]) have recently show that deep learning methods do not outperform classical baselines in many cases. Another issue I did not not see highlighted - the transformer + discretization approach might find it hard scale to multivariate time series (as the number of token to be predicted might expand by two orders of magnitude, in some cases). Although one of the benchmarks had 2 dimensions, it is not clear to me how this method would scale to standard benchmarks used by the deep learning time series AD community (SWaT, MSL, WADI etc.).

Clarity: The paper is well written, was a pleasure to read.

Significance: Due to the concerns mentioned above and the limited originality, I think this work is at risk of having low significance.

[1] Yan, Xudong, et al. "Learning semantic context from normal samples for unsupervised anomaly detection." AAAI'21
[2] Kim, Siwon, et al. "Towards a rigorous evaluation of time-series anomaly detection." AAAI'22.

---

> ### Author Response · Authors · 2022-08-02
> **Response**
>
> We are thankful to the reviewer for his/her helpful and informative comment. Below is the response to the particular point raised by the reviewer.
>
> >**(Weakness-Originality) The main premise of using inpainting as a task for reconstruction-based anomaly detection in not new. Using a transformer and the particular masking approach is an original contribution but a modest one.**
>
> One of our contributions is not only the proposal of a simple but effective transformer-based GAN framework, but also an effective pre-processing and embedding method in our framework. On top of that, we propose the original two-step masking that allows accurate anomaly detection by providing feedback within only two inferences. In case of SCADN [34], it requires multiple inferences for all masks in the mask pool to perform anomaly detection.
>
> [34] Yan, Xudong, et al. "Learning semantic context from normal samples for unsupervised anomaly detection." AAAI 2021
>
> >**(Weakness-Quality #1) The GAN ablation is somewhat worrisome as it appears to be useful only in a particular configuration and could be due to randomness in the data or a particular hyperparameter choice.**
>
> Using the same hyperparameters for NeurIPS-TS, Power-demand, 2D-gesture, and MIT-BIH, our model successfully converged in all experiments without any stability issue. Moreover, in Table 3, we conducted another experiments on different structures of the generator and the critic, and there were no stability issue too.
>
> >**(Weakness-Quality #2) The NeurIPS-TS datasets were generated by the authors, and are really easy. And the original datasets were much smaller than the ones generated here.**
>
> For the fair comparison, in the generated NeurIPS-TS dataset, we matched the number of training samples with the reported ones in Anomaly Transformer [11] as much as possible, i.e., 20,000 samples in Anomaly Transformer and 18,000 samples in AnoFormer.
>
> [11] Xu, Jiehui, et al. "Anomaly transformer: Time series anomaly detection with association discrepancy." ICLR 2022
>
> >**(Question #2, Weakness-Quality #3) Several papers (e.g. NeurIPS-TS cited in this paper or [a1]) have recently show that deep learning methods do not outperform classical baselines in many cases. But simple/classical baselines were not attempted. How do the classical benchmarks for various datasets perform?**
>
> Note that the original paper [38] compared the classical baselines with classic deep learning methods, e.g., simple RNN or GAN. In recent work [11], it already outperformed the classical baselines in all the benchmark datasets, which means deep learning methods show good performance in various datasets. We used the recent work [11] as one of our baselines.
>
> [11] Xu, Jiehui, et al. "Anomaly transformer: Time series anomaly detection with association discrepancy." ICLR 2022
>
> [38] Lai, Kwei-Herng, et al. "Revisiting time series outlier detection: Definitions and benchmarks." NeurIPS 2021
>
> [a1] Kim, Siwon, et al. "Towards a rigorous evaluation of time-series anomaly detection." AAAI 2022
>
>
>
> >**(Question #1, Weakness-Quality #4) Also the other 3 benchmarks are not standard. It is not clear to me how they were selected.**
>
> We chose the three benchmarks for the univariate case that are widely used in many recent papers [2, 7, 10, a2].
>
> [2] Zhou, Bin, et al. "BeatGAN: Anomalous Rhythm Detection using Adversarially Generated Time Series." IJCAI 2019
>
> [7] Shen, Lifeng, Zhuocong Li, and James Kwok. "Timeseries anomaly detection using temporal hierarchical one-class network." NeurIPS 2020
>
> [10] Shen, Lifeng, et al. "Time series anomaly detection with multiresolution ensemble decoding." AAAI 2021
>
> [a2] Boniol, Paul, and Themis Palpanas. "Series2graph: Graph-based subsequence anomaly detection for time series." VLDB 2020
>
> >**(Question #3, Weakness-Quality #5) How well does the method scale to multivariate TS (e.g. 100 dimensions)?**
>
> AnoFormer can be easily scaled up for the multivariate case. To process the multivariate dataset, we first made a single embedding vector by summing all embeddings from multi-channels. Since the dimension of the input embedding vector is the same as the univariate case, the following reconstruction process is exactly the same, and there is no additional computation cost.

---

> > ### Comment · Reviewer_73J5 · 2022-08-08
> > **Response**
> >
> > I thank the authors for their rebuttal. My responses:
> >
> > WO1: This is acknowledged, although to this reviewer these contributions seem of mild significance.
> > WQ1: While this is indeed good, still, convergence is not the whole story. In anomaly detection sensitivity to hyper-parameters is important as nearly always peaking into the test data is performed in hyper-parameter selection. The high sensitivity of AD to hyper-parameters is a cause for concern.
> > WQ2: Thank you for this clarification.
> > Q1: While the rebuttal presented.a few papers that use these benchmarks, these are certainly not the most common benchmarks and the reason for selecting them remains unclear.
> > Q2: The original paper used a much smaller dataset, so this isn't really comparable.
> > Q3: While this might be a valid strategy, it was not properly evaluated for more than two dimensions so it is speculative that this strategy should work.
> >
> > Overall, I do not see a reason to increase my score.

---

> > > ### Author Response · Authors · 2022-08-09
> > > **Response**
> > >
> > > Thank you for your feedback.
> > >
> > > >**WQ1: While this is indeed good, still, convergence is not the whole story. In anomaly detection sensitivity to hyper-parameters is important as nearly always peaking into the test data is performed in hyper-parameter selection. The high sensitivity of AD to hyper-parameters is a cause for concern.**
> > >
> > > In supplementary material Table 2, 3, and 4, we already conducted sensitivity experiments on the used hyperparameters (e.g. the quantization resolution, the stride of the sliding window, the number of masks in the pool, and the re-masking ratio). As reported in the tables, our model showed robust performances while changing the values of hyperparameters, and we selected the hyperparameters showing the highest performance. In the loss function, we set all the values of lambda as one without additional tuning, and showed stable performances.
> > >
> > > >**Q1: While the rebuttal presented a few papers that use these benchmarks, these are certainly not the most common benchmarks and the reason for selecting them remains unclear.**
> > >
> > > We will add the experimental results on other datasets, e.g. SWaT, MSL, WADI, etc., in the revision.
> > >
> > > >**Q2: The original paper used a much smaller dataset, so this isn't really comparable.**
> > >
> > > To the best of our knowledge, the original paper specified only the length of the sequence, i.e., the window size and the number of data samples are unknown. For this reason, we generated the dataset similar to the one of Anomaly Transformer [11]. We can compare the performances of the classical baselines reported in Anomaly Transformer [11].
> > >
> > > [11] Xu, Jiehui, et al. "Anomaly transformer: Time series anomaly detection with association discrepancy." ICLR 2022
> > >
> > > >**Q3: While this might be a valid strategy, it was not properly evaluated for more than two dimensions so it is speculative that this strategy should work.**
> > >
> > > We will conduct additional experiments on multi-dimensional datasets in the revision.

---

### Official Review · Reviewer_W6VQ · 2022-07-11

**Rating:** 4
**Confidence:** 5
**Soundness:** 3 good
**Presentation:** 2 fair
**Contribution:** 3 good

**Summary:**

The paper proposes a novel transformer-based GAN framework called AnoFormer for time series anomaly detection. In addition, the proposed model is enhanced by a two-step masking strategy consisting of 1) random masking and 2) entropy-based re-masking. The experimental results show the proposed AnoFormer achieves state-of-the-art performance on several benchmark datasets on time series anomaly detection.

**Questions:**

Why do we make the input signal discrete instead of continuous?

Why do we build the mask pool like this?

Why do we re-mask 50% of parts masked in Step 1?

Why the hyper-parameter $\epsilon$ is set to random? What is the distribution $P_{X'}$ in Line 186?

The performance of the paper is attractive, however, considering the novelty and the unclear elaboration, I vote for borderline reject.

**Strengths And Weaknesses:**

Strengths

+ The paper is easy to follow and well-organized.
+ The transformer-based GAN framework and the two-step masking strategy work well.
+ The performance gains are significant.

Weaknesses

- Lack of motivation. The paper only talks about how it is designed, while not elaborating clearly on why it is designed like this. E.g. Why do we make the input signal discrete instead of continuous? Why do we build the mask pool like this? Why do we re-mask 50% of parts masked in Step 1? Why the hyper-parameter $\epsilon$ is set to random?
- Lack of novelty and related work. Actually, the transformer-based GAN framework for time series is not novel and has been proposed in [1], which is not cited and discussed in this paper. The related work section should be more specific (e.g. time series related or time series anomaly detection related literature). The baselines, such as BeatGAN[2], TadGAN[3], RAMED[4], and Anomaly Transformer[5], should be discussed in this section.
- Some notations and presentations are confusing. E.g. $\epsilon$ in Line 185, how it is randomly sampled, with uniform random or normal random? And what is $P_{X'}$ in Line 186?
- The performance is good but codes are not provided.

[1] Adversarial Sparse Transformer for Time Series Forecasting. NeurIPS 2020

[2] BeatGAN: Anomalous Rhythm Detection using Adversarially Generated Time Series. IJCAI 2019

[3] TadGAN: Time Series Anomaly Detection Using Generative Adversarial Networks. IEEE International Conference on Big Data 2020

[4] Time Series Anomaly Detection with Multiresolution Ensemble Decoding. AAAI 2021

[5] Anomaly Transformer: Time Series Anomaly Detection with Association Discrepancy.  ICLR 2022

---

> ### Author Response · Authors · 2022-08-02
> **Response**
>
> We are thankful to the reviewer for his/her helpful and informative comment. Below is the response to the particular point raised by the reviewer.
>
> > **(Weakness #1, Question #1) Why do we make the input signal discrete instead of continuous?**
>
> We already mentioned the effectiveness of the proposed embedding method, i.e., discretization, in supplementary material D. As shown in Figure 2 of supplementary material, we confirmed that replacing the linear layer (for continuous signal) with the embedding table (for discrete signal) greatly improves the AUROC performance from 0.8493 to 0.9758. In other words, by transforming a continuous signal into a discrete signal, we can replace the reconstruction loss to the cross-entropy loss. Therefore, we change the regression problem to the easier classification problem, which predicts discrete class labels.
>
> > **(Weakness #1, Question #2) Why do we build the mask pool like this?**
>
> Instead of fully random masking, we designed a mask pool and randomly selected the mask within it to reduce the computation cost and excessive randomness. To this end, we predefined the masks in the mask pool that cover the different parts from each other, providing a complementary effect. As a result, in Table 3, the mask pool shows better performance than fully random masking with lower complexity.
>
>
> > **(Weakness #1, Question #3) Why do we re-mask 50% of parts masked in Step 1?**
>
> In supplementary material Table 4, we presented the performances per different re-masking ratio. We empirically found the re-masking ratio as 50%, which showed the best performance. In other words, 50% was enough to provide feedback on Step 1.
>
> > **(Weakness #1, #3, Question #4) Why the hyper-parameter $\epsilon$ is set to random? What is the distribution $P_{X′}$ in Line 186?**
>
> In Equation 6 in line 185, we adopted $\epsilon$ and $P_{X’}$ from WGAN-GP [36], where we sampled $\epsilon$ randomly from the uniform distribution on the interval (0, 1) to make the mixed signal $X’$, and $P_{X’}$ is the mixed distribution sampled uniformly along straight lines between pairs of points sampled from the data distribution $P_{\widetilde{X}}$ and the generator distribution $P_{\hat{X}}$.
>
> [36] Gulrajani, Ishaan, et al. "Improved training of wasserstein gans." NeurIPS 2017
>
>
> > **(Weakness #2) Lack of novelty. Actually, the transformer-based GAN framework for time series is not novel and has been proposed in [a1], which is not cited and discussed in this paper.**
>
> The referred Adversarial Sparse Transformer (AST) [a1] adopts an encoder-decoder based Sparse Transformer as the generator and a MLP-based discriminator for time series forecasting. On the other hand, AnoFormer configures both the generator and the discriminator using a simple transformer encoder for the anomaly detection task. In Table 2, we experimentally showed that the discriminator having the same architecture with the generator, i.e., transformer encoder, greatly improves the performance. To sum up, we propose a simple yet effective transformer-based GAN framework for the anomaly detection task using only the transformer encoder, along with an effective pre-processing and embedding method in our framework.
>
> [a1] Wu, Sifan, et al. "Adversarial sparse transformer for time series forecasting." NeurIPS 2020
>
> > **(Weakness #2) Lack of related work. The related work section should be more specific (e.g. time series related or time series anomaly detection related literature). The baselines, such as BeatGAN[2], TadGAN[8], RAMED[10], and Anomaly Transformer[11], should be discussed in this section.**
>
> Because of the page limit, in the related work section, we mainly describe the most related topics with our work, e.g., the transformer-based generative model and the masking technique. In case of baselines, we briefly introduced BeatGAN [2], TadGAN [8], RAMED [10], and Anomaly Transformer [11] in line 29 (1. Introduction) and line 200 (3. Experiments), but as per your comment, we will supplement the related work in the revision.
>
> [2] Zhou, Bin, et al. "BeatGAN: Anomalous Rhythm Detection using Adversarially Generated Time Series." IJCAI 2019
>
> [8] Geiger, Alexander, et al. “TadGAN: Time Series Anomaly Detection Using Generative Adversarial Networks.” IEEE International Conference on Big Data 2020
>
> [10] Shen, Lifeng, et al. "Time series anomaly detection with multiresolution ensemble decoding." AAAI 2021
>
> [11] Xu, Jiehui, et al. “Anomaly Transformer: Time Series Anomaly Detection with Association Discrepancy.” ICLR 2022
>
> > **(Weakness #4) The performance is good but codes are not provided.**
>
> We attach an anonymous github with the inference code (https://anonymous.4open.science/r/AnoFormer_NeurIPS_2022). We will release the training code too after the review.

---

> > ### Comment · Reviewer_W6VQ · 2022-08-07
> > **Reponses to Authors.**
> >
> > Thanks for the authors' detailed rebuttal. The authors almost address all my questions and concerns. Based on the originality and novelty (also mentioned by another reviewer), I would like to keep my score.

---

### Official Review · Reviewer_ZSRp · 2022-07-11

**Rating:** 5
**Confidence:** 4
**Soundness:** 2 fair
**Presentation:** 3 good
**Contribution:** 3 good

**Summary:**

The paper proposed a transformer-based generative model, named anoformer, to detect anomaly for time series in an unsupervised manner. Specifically, anoformer combined the Transformer model and the GAN model by adopting transformer-blocks as the generator and discriminator module of GAN. Besides, a two-step masking mechanism was designed to improve the model robustness. Through extensive experiments on 4 public datasets, including real-world and synthetic datasets, the superiority of anoformer was verified.

**Questions:**

please clarify the question below
* One sample in this paper is the entire time series or the subsequence of the time series? If it is subsequence, how do you do the sub-sampling.

* Whether the Anoformer can provide the anomaly detection in a real-time manner ? Please states the computing complexity and execution speed information.

* The formulation of the critic C is missing in the section 3.4, please further clarify it.

**Limitations:**

We have concerns on the following points:

* The experiment section only introduces the F1-based metrices, while F1-score highly depends on the threshold. Other metrics, e.g., AUC, the complexity, the running time, should be added.

* It is not very clear Anoformer provides the anomaly detection service on the data-point level or the time-series level? It will be great differences on performance and execution complexity.


**Strengths And Weaknesses:**

Strengths:
+ This paper is easy to follow with some well-designed figure to visualize the key modules of the Anoformer framework. In particular, each key module of Anoformer is elaborated on in following sections.
+ The introduction and the related work section provide a clear and brief introduction of the transformer-based generative models and the difference between the Anoformer and SOTA models.
+ The two-step mask is designed to avoid the generator just copying the input as the output. Specifically, the mask pool in the random mask step provides different views of the input; the exclusive mask in the re-mask step makes the model consider all parts of the input; the entropy-base mask in the re-mask step makes the model consider the uncertain part.
+ The extensive ablation experiment on different datasets verify the contribution of each key module of the Anoformer.


Weakness:
- Anoformer detects anomaly based on the entire input or the subsequence of the input. The length of the input time series impacts the delay of the anomaly detection service.
- Too many hyper-parameters are introduced in different sections. A summary table of these parameters should be added for a better review experience.
- The referred paper [30] was cited as both action recognition and text classification reference in line 81-82, while [30] only relates to action recognition.

---

> ### Author Response · Authors · 2022-08-02
> **Response**
>
> We are thankful to the reviewer for his/her helpful and informative comment. Below is the response to the particular point raised by the reviewer.
> > **(Question #1) One sample in this paper is the entire time series or the subsequence of the time series? If it is subsequence, how do you do the sub-sampling.**
>
> In this paper, we used the subsequence of the time series for all the experiments. For NeurIPS-TS, Power-demand, and 2D-gesture, we divided each time series into subsequences of the predefined length T, e.g. 100 for NeurIPS-TS, 512 for Power-demand, and 64 for 2D-gesture, followed by [10]. For MIT-BIH, we constituted a subsequence with a length T of 320 from the concatenation of 140 time ticks (about 0.4 sec) on the left and 180 time ticks (about 0.5 sec) on the right based on R-peaks, followed by [2].
>
> [2] Zhou, Bin, et al. "BeatGAN: Anomalous Rhythm Detection using Adversarially Generated Time Series." IJCAI 2019
>
> [10] Shen, Lifeng, et al. "Time series anomaly detection with multiresolution ensemble decoding." AAAI 2021
> > **(Weakness #1, Question #2) Whether the AnoFormer can provide the anomaly detection in a real-time manner? Please states the computing complexity and execution speed information.**
>
> We use the subsequence of the time series as the input. We measured execution time for each subsequence. AnoFormer takes 9.28 ms for NeurIPS-TS, 46.39 ms for MIT-BIH, 44.25 ms for Power-demand, and 8.26 ms for 2D-gesture. In the case of MIT-BIH, which has the longest execution time, it takes 1 second to measure one heartbeat and 46.39 ms to process one, so it is sufficient to handle the input signal in real-time by performing anomaly detection on the other signal while one signal is being an input.
> >**(Weakness #2) Too many hyper-parameters are introduced in different sections. A summary table of these parameters should be added for a better review experience.**
>
> We will gladly include the following [Table r1] in the revision:
> | Symbol | Description |
> |--|--|
> | $T$ | Length of the input signal |
> | $n$ | Channel of the input signal |
> | $K$ | Quantization resolution of the input signal |
> | $d$ | Input embedding dimension |
> | $\beta$ | Large constant (e.g. 1000) for soft-argmax operation |
> | $l_m$ | Length of a single mask section |
> | $r_m$ | Ratio of all mask parts in the entire length |
> | $s_m$ | Stride of the sliding window in the mask pool |
> | $n_m$ | The number of masks in the mask pool |
> | $\epsilon$ | A value of the uniformly random proportion of real/fake per input signal |
> | $\lambda$ | Scale factor of gradient penalty of WGAN-GP loss |
> | $\lambda_{rec}$ | Scale factor of reconstruction loss |
> | $\lambda_{adv}$ | Scale factor of adversarial loss |
> >**(Weakness #3) The referred paper [30] was cited as both action recognition and text classification reference in line 81-82, while [30] only relates to action recognition.**
>
> This is typo. We originally intended to refer [30] for action recognition and [a1] for text classification. We will correct the reference in the revision.
>
> [30] Cheng, Yi-Bin, et al. "Motion-transformer: self-supervised pre-training for skeleton-based action recognition." ACM 2021
>
> [a1] Moon, Seung Jun, et al. "Masker: Masked keyword regularization for reliable text classification." AAAI 2021
> >**(Question #3) The formulation of the critic $C$ is missing in the section 3.4, please further clarify it.**
>
> As explained in line 134, following the notation of WGAN-GP [36], we use the term critic instead of the discriminator. Equation 7 in line 184 represents the loss function of the critic $C$, which is the same as $\mathcal{L}_{adv}^c$ in Equation 10 in line 191.
>
> [36] Gulrajani, Ishaan, et al. "Improved training of wasserstein gans." NeurIPS 2017
> >**(Limitation #1) The experiment section only introduces the F1-based metrices, while F1-score highly depends on the threshold. Other metrics, e.g., AUC, the complexity, the running time, should be added.**
>
> In Table 1, 2, and 3, we already reported AUROC (Area Under the Receiver Operating Characteristic Curve, a.k.a. AUC) and AUPRC (Area Under the Precision-Recall Curve, a.k.a. PRC) along with F1-score for all experiments. In case of the complexity, since we construct the generator and the critic with pure transformer encoder, the complexity per layer is $O\left(T^2\cdot d\right)$ from [14]. Lastly, in the running time, we mentioned in (Weakness #1, Question #2).
>
> [14] Vaswani, Ashish, et al. "Attention is all you need." NeurIPS 2017
> >**(Limitation #2) It is not very clear AnoFormer provides the anomaly detection service on the data-point level or the time-series level? It will be great differences on performance and execution complexity.**
>
> As explained in line 100 (3.1 Problem Definition), AnoFormer provides the anomaly detection service on the time-series level, which means it can detect whether the signal is normal or not based on the reconstruction errors between the given signal and the generated signal.

---

### Meta-Review · Area_Chair_shnB · 2022-08-27

**Recommendation:** Reject
**Confidence:** Certain

**Metareview:**

This paper proposes a transformer-based GAN method and a two-step masking mechanism for time series anomaly detection. The proposed method is demonstrated on a variety of datasets.

After rebuttals, both Reviewer 73J5 and Reviewer W6VQ remained negative. The main concern is the novelty and significance of the proposed method.

**Award:**

No

---

### Decision · Program_Chairs · 2022-09-14

Reject